# Feasibility and Effectiveness of a Preventive Care Program during the Compound Humanitarian Crisis and COVID-19 Pandemic in Venezuela

**DOI:** 10.3390/nu14050939

**Published:** 2022-02-23

**Authors:** Juan P. González-Rivas, María M. Infante-García, Ramfis Nieto-Martinez, Jeffrey I. Mechanick, Goodarz Danaei

**Affiliations:** 1Departments of Global Health and Population and Epidemiology, Harvard T.H. Chan School of Public Health, Harvard University, Boston, MA 02115, USA; nietoramfis@gmail.com (R.N.-M.); gdanaei@hsph.harvard.edu (G.D.); 2International Clinical Research Center (ICRC), St. Anne’s University Hospital (FNUSA) Brno, Pekařská 53, 656 91 Brno, Czech Republic; maria.garcia@fnusa.cz; 3Foundation for Clinic, Public Health, and Epidemiological Research of Venezuela (FISPEVEN), Barquisimeto 3001, Venezuela; 4The Marie-Josée and Henry R. Kravis Center for Cardiovascular Health at Mount Sinai Heart, Icahn School of Medicine at Mount Sinai, New York, NY 10028, USA; jeffreymechanick@gmail.com; 5Division of Endocrinology, Diabetes and Bone, Icahn School of Medicine at Mount Sinai, New York, NY 10028, USA

**Keywords:** type 2 diabetes mellitus, humanitarian crisis, primary prevention, COVID-19

## Abstract

Effective preventive care programs are urgently needed during humanitarian crises, as has been especially obvious during the COVID-19 pandemic. A pragmatic trial was designed: hybridized intervention (Diabetes Prevention Program [DPP] + medical nutrition therapy + liquid diet [LD]; LD group) vs. DPP only (DPP group). The participants were adults who were overweight/obese and at high risk of type 2 diabetes mellitus (T2DM). The LD consisted of a “homemade” milk- and fruit-juice-based beverage. Pandemic restrictions delayed the program by nine months, tripled the amount of time required for screening, and reduced the total sample to 60%. Eventually, 127 participants were randomized, and 94/127 participants (74.0%) completed the first phase. Participant dropout was influenced by migration, COVID-19 symptoms, education level, and socioeconomic status. In two months, the LD group lost 2.9 kg (*p* < 0.001) and the DPP group, 2.2 kg (*p* < 0.001) (between-group *p* = 0.170), with improvements in their cardiometabolic risk factors. At this stage, the DPP was shown to be feasible and effective, demonstrating weight loss with the improvement of cardiometabolic risk factors in a primary setting in Venezuela, a middle-income country with a chronic humanitarian crisis, during the COVID-19 pandemic.

## 1. Introduction

Type 2 diabetes mellitus (T2DM) is a global public health problem that is disproportionally affecting low- and middle-income countries [1]. In Venezuela, the diabetes mortality rate has increased by 107.6% in the last two decades, representing the fourth most common cause of death in 2019, and the third most common cause of disability-adjusted life years (DALYs) [2]. This disease burden largely results from the increasing prevalence of prediabetes and T2DM, rising from 14.4% and 8.0% in 2006–2010 to 34.9% and 12.3% in 2014–2017, respectively, as well as the collapse of the healthcare system because of the chronic humanitarian crisis [3,4].

In subjects with abnormal adiposity, insulin resistance, and/or prediabetes, lifestyle intervention programs targeting weight loss can prevent or delay T2DM incidence. The US Diabetes Prevention Program (DPP) and the Finnish Diabetes Prevention Study (DPS) are two primary prevention studies that showed a reduction in the incidence of T2DM by 58% compared to standard care, reporting weight reductions of 5% to 7% [5,6]. Since their publication two decades ago, many pragmatic clinical trials (evaluating program effectiveness in “real-world” and not necessarily optimal settings) have tried to replicate the benefits observed in these studies, but with less success. Specifically, a meta-analysis of 36 pragmatic primary prevention trials showed a reduction in T2DM incidence by 26%, half that found in the DPP/DPS studies, and an average weight loss of 1.5 kg in the intervention group compared to standard care [7].

The DPP demonstrated that for each kilogram of weight loss, the incidence of T2DM was reduced by 16% [8], focusing strategies regarding T2DM prevention toward weight change. Total diet replacement with low-energy liquid or solid diets (<1200 kcal/day) is an effective non-surgical, non-pharmacological strategy to reduce weight [9]. In the PREVIEW study (PREVention of diabetes through lifestyle intervention and population studies In Europe and around the World), 2224 participants with prediabetes and obesity, drawn from 6 countries in Europe and Australia/New Zealand, received a low-energy diet for 8 weeks and achieved a mean weight loss of 10.7 ± 0.4 kg [10]. Moreover, 83.5% achieved the target of ≥8% weight reduction, producing a three-year expected T2DM incidence of only 3.1%, less than one-third of the predicted incidence (10.5–15.8%) [11], independent of the intervention used post-weight loss. The Diabetes Remission Clinical Trial (DIRECT) study targeted T2DM remission and included 306 participants who were <6 years post-diagnosis, drawn from primary healthcare centers in the UK [12]. Participants received total diet replacement with a liquid diet (LD), versus standard care, for 3–5 months (825–853 kcal/day). After a year of intervention, the LD group lost 10 kg, compared with 1 kg in the standard-care group (*p* < 0.001). In this study, T2DM remission varied with the amount of weight loss: remission was not achieved in those who gained weight; remission was observed in 7% who achieved 0–5 kg weight loss; 34% reported 5–10 kg weight loss; 57% reported 10–15 kg weight loss; 86% reported a loss of 15 kg or more [12].

These studies highlight effective strategies for weight loss in those with T2DM who are in primary healthcare settings. However, these studies were implemented in high-income countries, where LD formulas were provided free of charge during the period of intervention. A critical challenge to the effectiveness of these strategies is their implementation in resource-constrained settings. This is addressed by a pragmatic clinical trial, published as: “Hybridized Three Steps (HITS) Intervention to Prevent Diabetes in Venezuela: HITS Diabetes with Prevention, An EVESCAM Lifestyle Intervention Study”. This paper presents the results of the first phase of this trial, assessing the implementation process of feasibility and fidelity, and the effectiveness in weight loss of using a hybridized approach with LD, medical nutrition therapy, and an adapted DPP vs. using an adapted DPP alone. This paper also takes into account the impact of various social determinants of health and, furthermore, describes how the hurdles presented by the COVID-19 pandemic were overcome for successful study implementation.

## 2. Materials and Methods

This pragmatic clinical trial compares the weight loss achieved with two lifestyle intervention programs in a community health center (CHC): the LD group—a 3-step hybrid lifestyle intervention (step 1—inducing rapid weight loss with total LD replacement, step 2—medical nutrition therapy, and step 3—the adaptation of the DPP); and the DPP group—only applying an adaptation of the DPP. The study design includes two phases: Phase 1 corresponds to the first two months of intervention (LD group step 1 vs. DPP group) and Phase 2 corresponds to the end of the sixth month when both groups completed the intervention.

### 2.1. Study Site

Venezuelan communities have “communal houses”, which are community CHCs where different activities (e.g., dancing, aerobics, yoga, karate, educative training, and meetings) are conducted, where the population is medically attended by a primary care physician. A CHC in the city of Merida was selected as the site of intervention. Merida is in the Andes region, with 508,988 inhabitants in 2018. The city is organized into “Communal Councils”, representing a location or small sector; two or more Communal Councils create a “Commonwealth”. The CHC selected for the study is in a Commonwealth, middle-class urban area. Community leaders of this Commonwealth are regularly involved in implementing healthy activities in their neighborhoods.

### 2.2. Inclusion and Exclusion Criteria

The inclusion criteria were: adults ≥ 20 years old, no history of T2DM, body mass index (BMI) ≥ 25 kg/m^2^, and high risk for T2DM were included. The exclusion criteria were: self-reported ischemic heart disease (acute myocardial infarction, stable angina, or unstable angina), stroke, use of anticoagulants, severe renal failure, heart failure, inability to engage in moderate-intensity physical activity, inability to attend most sessions, pregnancy or intention to become pregnant during the next six months, cancer or chemotherapy, the use of medications that affect weight, or plans for migration in the next year, as well as physical examination findings by the general practitioner of peripheral vascular disease, clinical osteoarthritis of the knees, ankles or hips, BMI ≥ 40 kg/m^2^, systolic blood pressure ≥ 180 mmHg or diastolic blood pressure ≥ 110 mmHg, or a fasting plasma glucose of ≥126 mg/dL or a glucose level after the oral glucose tolerance test (OGTT) of ≥200 mg/dL. To avoid the risk of contamination bias, only one member per family was enrolled in the study.

### 2.3. Sampling and Recruitment

Community members were invited to attend a medical screening in the CHC. Those at high risk of T2DM were identified using the Latin America Finnish diabetes risk score (LA-FINRISC) [13], a non-invasive tool that includes the variables of age, BMI, waist circumference, physical activity, daily consumption of fruits and vegetables, history of hyperglycemia, history of antihypertensive drug treatment, and family history of diabetes, assigning a score ranging from 0 to 26 points. In Venezuelan adults, a score of more than 9 points has a sensitivity of 71.4% and specificity of 65.4% to detect impaired glucose tolerance [14]. Those with an LA-FINDRISC of ≥10 points and who complied with the inclusion criteria were invited for laboratory testing and medical evaluation. Those found to have mild hyperglycemia in the prediabetes range or normoglycemia were invited to participate in the study.

Based on unpublished pilot study results in the community, the study was powered to detect a two-fold reduction of weight loss in the LD group (mean = 5.0 kg, standard deviation = 4.9) compared with the DPP group (mean = 2.4 kg, standard deviation = 1.9). By assigning a beta-error of 0.2 and alpha-error of 0.01, the sample size required for each group was 50; allowing for a 20% drop-out rate based on the pilot study and stipulating an additional 30% in the number of participants to ensure the representativeness of the sample and power, the sample size was incremented to 78 participants in each group, with a total sample size of 156.

### 2.4. Transculturalization

#### 2.4.1. The Diabetes Prevention Program (DPP)

The DPP Group Lifestyle Balance (GLB) core curriculum was modified from the original DPP and was accessed online in both English and Spanish [15]. The transcultural diabetes nutrition algorithm (tDNA) process [16,17] was implemented via the following steps to transculturalize DPP-GLB content: Identify the target population: Venezuelan adults.Identify the research/clinical question: How can evidence-based preventive care strategies be applied to reduce the T2DM burden?Identify, recruit, and involve a team of experts in the source (DPP) and target populations (Venezuelan): a group of Venezuelan experts (four diabetologists, one primary care physician, and two community members) are trained to culturally adapt DPP-GLB content for study in Venezuela.Identify, codify, and organize culturally adapted DPP-GLB content using a pragmatic framework: here, DPP-GLB content was interpreted using the ecological validity model (EVM) [18].

Using an interactive process among team members, discrepancies between content and customs were identified and addressed in the EVM framework (e.g., culinary recipes were adapted to suit the poor socio-economic status in Venezuela). Subsequently, a new, culturally adapted T2DM prevention curriculum for Venezuelan adults to prevent T2DM development and progression was created and made ready for implementation and study. Healthy food options, goal-setting, and problem-solving were consistent attributes of this culturally adapted program.

#### 2.4.2. Focus on the Low-Energy Liquid Diet

Total LD replacement with a low-energy beverage and structured food reintroduction is a feasible and acceptable strategy to lose and maintain weight [9]. Specifically, a “homemade” milk- and fruit juice-based beverage, similar in composition to the nutritional formulas provided in the DiRECT study [12] (811 kcal, 64 g protein, 132 g carbohydrate, and 6 g fat per day) was recommended to participants. Recipes and preparation techniques were culturally adapted to the population, using locally available low-cost foods. The palatability of the LD was tested and approved by members of the research team and the local community.

### 2.5. Intervention

All participants received guideline-directed management by a physician for their cardiometabolic health. Healthy lifestyle recommendations comprised healthy dietary patterns, increased moderate and vigorous physical activity to reach 150 min weeks, and 7% of weight loss.

#### 2.5.1. Standard Care (DPP Group)

A trained coach provided group-based DPP-GLB content in 16-sessions over 6-months. The DPP has two primary goals: achieving at least 7% weight loss and engaging in 150 min per week of moderate or vigorous physical activity. Participants were introduced to self-monitoring of weight and management of the amount and frequency of food consumption, with an emphasis on the reduction of calories coming from fats. Printed material based on DPP-GLB content was provided without cost to participants in each session.

#### 2.5.2. Hybridized Intervention (LD Group)

Participants in the LD group received a hybridized intervention in two phases:Phase 1: total LD replacement with a low-energy beverage for two months, followed by structured food reintroduction for two weeks, supervised by a nutritionist.Phase 2: medical nutritional therapy using a locally adapted tDNA Toolkit, supervised by a nutritionist. Standard care with DPP-GLB content was provided throughout this period, supervised by a coach.

This hybridized intervention had similar goals to the standard care group: achieving at least 7% weight loss and engaging in 150 min per week of moderate or vigorous physical activity. If the participant relapsed (i.e., a weight regain of >2 kg) after the first two months of phase 1 (during which time, only an LD was consumed), then the LD was re-introduced to replace one or two main meals per day for 4 weeks.

### 2.6. Impact of the COVID-19 Pandemic

Originally, the intervention was planned to take place in the CHC, in group sessions with 15 to 20 participants. However, to avoid COVID-19 infection, the number of groups was duplicated to reduce group sizes for social distancing, so that there were only 7 to 10 participants per session. Likewise, all group and some individual indoor activities were replaced by outdoor activities. Physical activities requiring contact among participants were also avoided.

### 2.7. Data Collection

A trained team collected the data in the CHC via face-to-face interviews. Questionnaires included sociodemographic characteristics, family and personal medical history, behavioral risk factors, stress, and mental health. Blood pressure was measured twice in the right arm, supported at heart level in a sitting position after five minutes of rest, with an oscillometric sphygmomanometer (Omron HEM-705C Pint^®^, Omron Healthcare Co., Kyoto/Japan) [19]. Weight and body composition analyses were performed using bioelectric impedance, with the lightest possible clothes, without shoes, using a calibrated scale (Tanita UM-081^®^, Tokyo, Japan). Height was measured using a portable stadiometer (Seca 206^®^ Seca GmbH & Co., Hamburg, Germany). The waist circumference was measured twice at the end of expiration, with a metric tape at the iliac crest in a horizontal plane with the floor; the average value was used. Blood samples were collected according to a standardized protocol after at least 8 h of fasting. The 2-h OGTT, using a 300 mL test solution containing 75 g anhydrous glucose, was performed. Samples were centrifuged, frozen, and analyzed in a local laboratory. Blood tests included total cholesterol, triglycerides, and HDL-c. LDL-c was calculated using Friedewald’s formula.

### 2.8. Variables Definition

Feasibility was defined using the administrative data of recruitment and retention and was presented as the percentage of participants that completed the intervention [20]. Fidelity was defined using administrative data about the number of encounters, and average and distribution of attendance, according to the number of sessions. The subjective socioeconomic status (SES) of each individual was determined using the MacArthur sociodemographic questionnaire—participants ranked themselves by indicating their place on a 10-rung ladder in relation to others. This questionnaire also attempts to capture the psychological aspects of social status by asking participants to think about their education, job, and income status relative to other participants; higher scores indicate a higher subjective SES [21]. The EQ-5D self-rated questionnaire includes a visual analog scale, which records the respondent’s self-rated health status on a graded scale (0–100), with higher scores for higher self-perceived quality of life [22]. Prediabetes was defined as a fasting plasma glucose of ≥100 and <126 mg/dL, or ≥140 mg/dL and ≤199 mg/dL 2-h after the OGTT [23]. Hypertension was defined as systolic blood pressure of ≥140 mmHg or diastolic blood pressure of ≥90 mmHg, or a self-report of hypertension or the use of antihypertensive medication [24]. Body mass index was calculated as weight in kilograms/height in m^2^. High cholesterol was defined as total cholesterol ≥ 200 mg/dL. High triglycerides were defined as a triglyceride level of ≥150 mg/dL.

### 2.9. Data Analysis

Analyses were performed using SPSS 20 software (IBM Corp., released 2011; Armonk, NY, USA). Fidelity was analyzed as the mean and standard deviation of sessions attended and was assessed using Student’s *t*-test. The distribution of attendance was presented in terms of number and proportion and was assessed using the chi-square test. The biological characteristics of the participants had a normal distribution (evaluated with the Kolmogorov–Smirnov test) and were summarized using a mean and standard deviation, and their difference at the baseline evaluation was assessed using Student’s *t*-test. The frequency of risk factors and social determinants of health at baseline were presented as percentages and were evaluated using the chi-square test. The mean of change of biological characteristics, before and after the intervention, was assessed using a linear regression model, adjusting each variable by their baseline value. The association between the number of sessions attended and the weight change was assessed using Spearman’s rank correlation, and the R and R2 linear values were presented. The number of sessions attended was categorized as low = 0–2; intermediate = 3–5; high = 6–8; the changes of weight in each category were presented as mean and standard deviations, the difference within each group of interventions (LD and DPP) was analyzed using an ANOVA. A sub-analysis of the difference in weight change between the intervention groups, including only those participants with intermediate and high attendance, was implemented using a two-way ANOVA, adjusting by the baseline weight.

### 2.10. Ethics and Clinical Trial

The study protocol complied with the Helsinki declaration and all participants signed the informed consent. The study was approved by the Ethics Committee of the Universidad Centroccidental Lisandro Alvarado, Barquisimeto, Venezuela, number CBDCS-03.2020, and is registered in ClinicalTrials.gov, no. NCT04927871.

## 3. Results

The flowchart for the design of the study is presented in Figure 1.

### 3.1. Feasibility

#### Overcoming the COVID-19 Pandemic Restrictions

The case-finding of potential participants was planned to be implemented in the community from March to April 2020, with evaluation sessions held with multiple attendees at the weekends. Based on prior experience with the community, about 80 participants were expected to attend per day, approximating 1000 participants in three months, with the intent to enroll the required sample size. However, in Venezuela, COVID-19 restrictions commenced in the fourth week of February and, therefore, evaluations were postponed for 9 months. In compliance with the COVID-19 regulations, the evaluations eventually started in November 2020 with a modified approach. Team members contacted community leaders of each building or small neighborhood and coordinated the evaluations in the surrounding areas of the building or outside the CHC, seeing on average 6 to 8 potential participants per day, 5 days per week. This strategy was implemented intermittently each week because the Venezuelan governmental restrictions for the pandemic were based on a “flexible week”, where the population could perform regular activities for almost the entire day, and a “radical week”, where the population was not allowed to leave their homes. The evaluation process for the study ended in April 2021. A total of 629 participants were evaluated, about 60% of the expected number, and 328/629 (52.1%) were invited for laboratory evaluation, of whom 148/328 (45.1%) accepted (Table 1). Of these, 127/148 (85.8%) were selected and were randomly assigned to participate in either group, with 64/127 (50.4%) in the LD group and 63/127 (49.6%) in the DPP group.

### 3.2. Dropout Rates

Phase 1 was completed with the evaluation of the participants at the second month of the intervention, after the LD phase ended and the first eight sessions of the DPP occurred. The assessment was completed by 94/127 participants (74.0%), with a dropout rate of 18/64 (28.1%) in the LD group and 15/63 (23.8%) in the DPP group, for a total dropout rate of 33/127 (26.0%) in Phase 1. In those who dropped out, 8/18 (44.4%) of the LD group “didn’t like the intervention”, while 2/18 (11.1%) developed COVID-19-related symptoms and missed the second evaluation. Despite the fact that an intention to migrate within the next year was an exclusion criterion, 7/15 (46.6%) of the participants dropping out of the DPP group migrated in the first 2 months. There was a loss of contact with 7 participants in each group (Table 1). Overall, those who dropped out reported a higher quality of life score (EQ-5D), lower education level, and lower self-perceived SES (Table 2). No biological differences correlated with dropout rates.

### 3.3. Fidelity

The average number of sessions attended was slightly lower in the LD group (4.8 ± 2.9 vs. 5.7 ± 1.7 in the only DPP group; *p* = 0.071). In total, 12/46 (26.1%) participants in the LD Group attended only one session, compared to 0 in the DPP group (*p* = 0.001) (Figure 2). Most of the participants in both groups (LD group, 16/46 (34.8%) and DPP group, 34/48 (70.8%)) attended 4 to 7 sessions out of a total of 8.

### 3.4. Participant Characteristics

Baseline characteristics were similar between groups, except for the prevalence of hypertension, which was higher in the DPP group (65.1%) than the LD group (43.8%) (*p* = 0.013). In total, 127 participants were included, with a mean age of 52.1 ± 12.0 years, BMI 30.2 ± 3.7 kg/m^2^, systolic blood pressure 127.2 ± 18.3 mmHg, diastolic blood pressure 74.2 ± 9.7 mmHg, fasting blood glucose 86.3 ± 7.9 mg/dL, and 2-h blood glucose level after the OGTT of 92.3 ± 24.4 (Table 3). Of these, 68.5% of the participants had a university degree, 9.4% were self-described as unemployed, and the mean self-perceived SES was 5.5 ± 1.5 points (ranging from 0 to 10). In total, 54.3% of the participants described their self-perceived personal financial situation as “regular”.

### 3.5. Cardiometabolic Risk Factors

Participants in both groups showed improvements in cardiometabolic risk factors after 2 months (Table 4). The LD and DPP groups demonstrated significant reductions in weight (2.9 kg and 2.2 kg), waist circumference (3.2 cm and 1.8 cm), body fat percent (1.4% and 1.9%), and systolic blood pressure (10.5 mmHg and 8.4 mmHg), respectively. Additionally, participants in the LD group showed reductions in BMI (1.2 k/m^2^), and those in the DPP group showed reductions in diastolic blood pressure (1.8 mmHg). The fasting blood glucose, which was quite low in both groups (87 mg/dL) at baseline, remained similar after 2 months. The OGTT and serum lipid profiles were not evaluated in Phase 1.

When changes in the cardiometabolic risks were compared between the groups, there were no significant differences. The LD group lost 2.9 kg compared to 2.2 kg in the DPP group (*p* = 0.170) (Table 4). The systolic blood pressure decreased by 10.5 mmHg in the LD group and by 8.4 mmHg in the DPP group, even though antihypertensive regimens remained the same in both groups, suggesting that the change was not related to medical treatment but rather due to the lifestyle changes. The number of sessions attended in the LD group correlated strongly with the weight change (R = −0.728; *p* < 0.001; R^2^ = 0.464), but not with the DPP group alone (R = −0.323; *p* = 0.055; R^2^ = 0.075) (Figure 3). 

In the LD group, those with low attendance (*n* = 11) gained 0.24 kg, intermediate attendance (*n* = 7) lost 2.43 kg, and high attendance (*n* = 24) lost 4.55 kg (*p* < 0.001) (Figure 4). In the DPP group, those with low attendance (*n* = 1) gained 1.3 kg, intermediate attendance (*n* = 18) lost 1.87 kg, and high attendance (*n* = 28) lost 2.53 kg (*p* = 0.292). When participants with low attendance were excluded from the analysis, and only those with intermediate to high attendance were included, the LD group lost 1.80 kg more than the DPP group (LD group = 4.06 ± 2.97 kg vs. DPP group = 2.27 ± 2.60 kg; *p* = 0.004).

## 4. Discussion

The implementation of Phase 1 of a study program comparing a transculturalized hybridized approach to T2DM preventive care in Venezuela that includes LD, the structured re-introduction of food, medical nutrition therapy, and DPP strategies, vs. an active comparator group using the DPP strategies alone was demonstrated to be feasible. The dropout rate of 26% was primarily influenced by low adoption of the LD, migration, a low education level, and a lower SES. In those participants who completed Phase 1, there was significant weight loss and an improvement in cardiometabolic risk factors. Weight loss was similar between intervention groups but was highly influenced by low attendance rates in the LD group. When only those with intermediate to high attendance were compared, the LD group lost more weight than the DPP group. Both weight loss and lifestyle changes were associated with large reductions in systolic blood pressure in both groups. Notably, the COVID-19 pandemic-imposed restrictions limited the possibilities of implementing much larger evaluation sessions; however, adopting recommended biosecurity measures and new logistical strategies allowed the successful implementation of this program but on a smaller scale.

The implementation of lifestyle intervention programs in low- and middle-income countries face local challenges that require feasibility studies. In the case of Venezuela, the country is facing a chronic humanitarian crisis, characterized by the collapse of the economy, the healthcare system, and basic services, coupled with an increase in violence [25]. As a consequence, 5.4 million Venezuelans migrated and left the country from 2014 to 2020 [26]. In this study, migration strongly influenced the study dropout rates. However, dropout is a common limitation in the implementation of preventive programs in clinical practice settings. For instance, in Colombia, the Demo Juan study was a T2DM prevention project involving 772 adults at high risk of T2DM, randomly allocated to three groups: standard care vs. the T2DM prevention program; starting with nutritional intervention vs. the T2DM prevention program; starting with physical activity intervention, for 24 months [27]. The primary outcomes were conversion back to normoglycemia or conversion to T2DM. The dropout rate of the study was 50.5%, far more than in the present study. Moreover, dropout rates in the Ontario Primary Care DPP in Canada (*n* = 1916) were 26.8%, 46.8%, and 63.0% at 3, 6, and 9 months, respectively [28]. In the case of the US DPP, the dropout rate increased progressively until it reached 68.1% at 11 months, and this figure was higher in younger age groups and among minority races/ethnicities [29]. In contrast, in the Kerala DPP study in India, 1007 adults were randomly assigned to receive either the DPP or standard care, with a dropout rate of less than 5% [30]. In the present study, low education and low SES were associated with higher dropout rates. Participants with these drawbacks may be more exposed to chronic stressors than those with a university education and high SES [30]; therefore, they need more resources to overcome the daily challenges of a stressful environment [30].

Despite an effort to culturally adapt the DPP and the LD to the Venezuelan population, the results here show that the LD was not sufficiently adopted to make the intervention effective at this study phase. The idea of the LD was derived from the DiRECT study in the UK, where total LD replacement was associated with large degrees of obesity [12]. The benefit of a low-calorie diet was also observed in the PREVIEW study in Europe and Australia/New Zealand [10]. However, in the DiRECT and PREVIEW studies, the diet replacements were provided free of charge to participants, the LD consisting of commercial nutritional formulas with instant preparation and known nutritional composition, reducing the possibility of modification by the participants. This strategy is not feasible during a humanitarian crisis. Instead, a culturally adapted homemade option was recommended in the present study, but this still represented a higher level of task complexity for the participants. Future strategies should be implemented using more easily affordable and culturally adapted recipes. In the Venezuelan case, a low-caloric diet with soups and salads that incorporate local tastes and culinary techniques would be most suitable. In Venezuela, 8.2% and 66.8% of the population eat soup daily and weekly, respectively, and 31.0% and 57% eat salads daily and weekly, respectively [31]. There are also popular healthy soups that are regularly consumed at weekends, such as “Sancocho”, a soup made with vegetables and beef or chicken [16].

The limitations of the present study relate to how and why the participants may have modified LD compositions, due to difficulties sourcing ingredients because of a lack of transportation, and the increasing cost of fruits and other components of the LD. However, since these are a consequence of infrastructural challenges, not only from the COVID-19 pandemic but also the humanitarian crises, the results of this study have greater pragmatic relevance. The loss of contact with those who dropped out meant that to ascertain the reasons is another limitation, as well as the use of a portable weight balance to estimate the body fat percentage and the inability to estimate visceral fat adiposity. The strengths of the present study relate to the impact of specific components of the humanitarian crisis, such as migration, and other social determinants of health as part of the feasibility analysis. In addition, this study provides insights about biosecurity measures during the COVID-19 pandemic and their impact on prevention study design.

## 5. Conclusions

In conclusion, a pragmatic preventive care program for T2DM can be implemented in a primary setting of Venezuela, a middle-income country with a chronic humanitarian crisis, in the throes of the COVID-19 pandemic. The dropout rate was influenced by the use of an LD, a massive migration rate, and various social determinants of health. Lifestyle interventions lead to significant weight loss, with an improvement of cardiometabolic risk factors, especially a large reduction in systolic blood pressure. However, over the short term (about 2 months), incorporating a culturally adapted LD confers no benefit over implementing the DPP alone. Pragmatic primary and secondary prevention programs are desperately needed to actually improve cardiometabolic health at the population level. Implementation strategies can only succeed when based on pragmatic clinical trials that incorporate transcultural factors under far from optimal circumstances.

## Figures and Tables

**Figure 1 nutrients-14-00939-f001:**
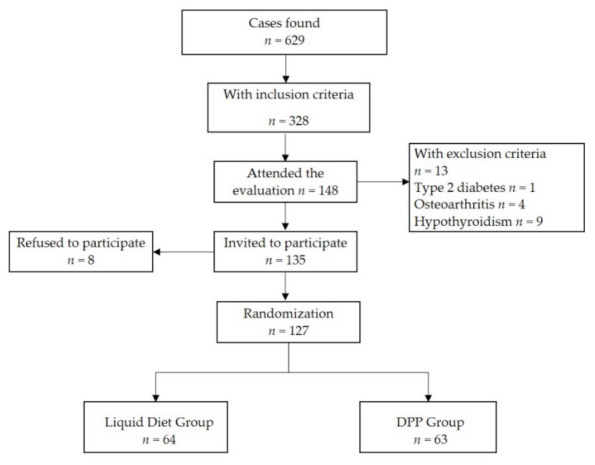
Flowchart of the study design.

**Figure 2 nutrients-14-00939-f002:**
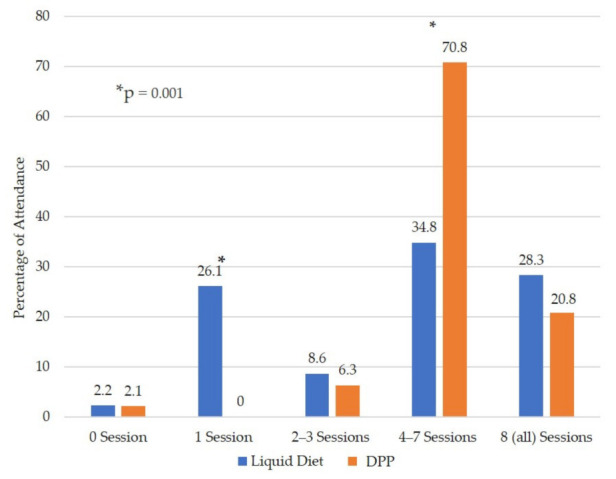
Fidelity of those who completed the second evaluation. * Significant difference between groups (*p* = 0.001).

**Figure 3 nutrients-14-00939-f003:**
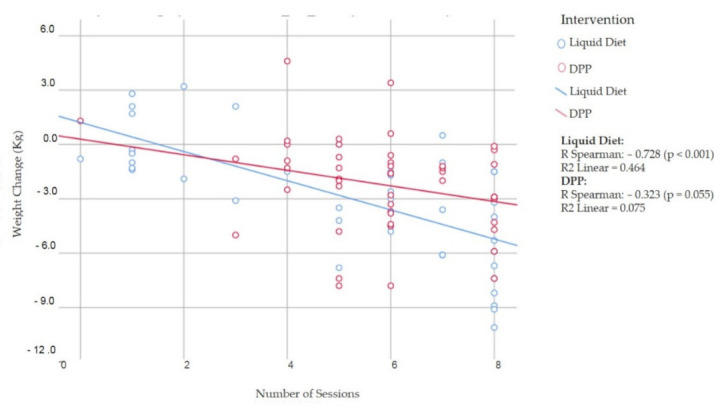
Correlation between weight change and the number of sessions attended.

**Figure 4 nutrients-14-00939-f004:**
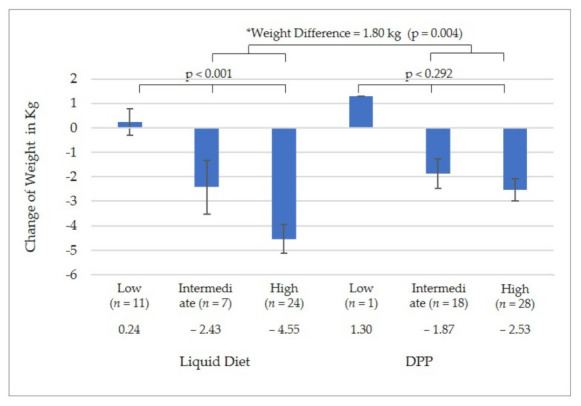
Change in weight by level of attendance. * Weight difference between groups excluding the participants with low attendance.

**Table 1 nutrients-14-00939-t001:** Feasibility assessment in the first phase of the program.

	*n*	%
**Feasibility**		
Cases found	629	100
Invited to evaluation with inclusion criteria (% of those screened)	328	52.1
Attended the evaluation (% of those invited)	148	45.1
Selected to participate (% of those evaluated)	135	91.2
**Groups**	**Liquid Diet**	**Only DPP**
Started *n* (%)	64	63
Completed the second-month evaluation *n* (%)	46 (71.8%)	48 (76.1%)
**Dropout Rates**	18 (28.1%)	15 (23.8%)
**Reasons for dropping out**	18	15
Migration	0 (0.0%)	7 (46.6%)
“Didn’t like the intervention”	8 (44.4%)	0 (0.0%)
COVID-19-related symptoms and so missed the second evaluation	2 (11.1%)	0 (0.0%)
Pregnancy	0 (0.0%)	1 (6.6%)
Economic reasons	1 (5.5%)	0 (0.0%)
Unknown (loss of contact)	7 (38.8%)	7 (46.6%)

**Table 2 nutrients-14-00939-t002:** Characteristics of the participants who completed or dropped out of the first phase.

	Completed	Dropout Rate	*p*-Value
*n*	94	33	
Women *n* (%)	71 (75.5%)	20 (60.6%)	0.102
Age (years)	52.4 ± 12.1	51.3 ± 11.9	0.675
Height (cm)	159.4 ± 7.6	162.1 ± 9.9	0.111
Weight (kg)	77.2 ± 12.2	79.8 ± 16.6	0.347
Body mass index (kg/m^2^)	30.3 ± 3.6	30.1 ± 3.9	0.867
Waist circumference (cm)	103.9 ± 8.1	106.3 ± 10.4	0.179
Fat percentage (%)	40.9 ± 7.9	38.4 ± 7.2	0.108
Systolic blood pressure (mmHg)	127.9 ± 18.4	125.2 ± 18.2	0.473
Diastolic blood pressure (mmHg)	74.3 ± 8.9	74.0 ± 12.0	0.896
Fasting blood glucose (mg/dL)	86.7 ± 7.6	84.9 ± 8.4	0.247
2-h blood glucose (mg/dL)	94.1 ± 24.5	87.1 ± 23.4	0.157
Total cholesterol (mg/dL)	182.0 ± 45.5	175.8 ± 34.8	0.477
LDL-c (mg/dL)	110.6 ± 35.1	107.0 ± 25.2	0.594
VLDL-c (mg/dL)	25.5 ± 11.4	26.5 ± 9.7	0.660
HDL-c (mg/dL)	41.5 ± 8.2	40.0 ± 7.6	0.375
Triglycerides (mg/dL)	137.5 ± 97.9	145.3 ± 86.7	0.686
EQ-5D	68.1 ± 15.7	75.4 ± 15.5	0.022
**Social Determinants**			
**Education Level**			
Primary	5 (5.3%)	8 (24.2%)	0.008
Secondary	22 (23.4%)	5 (15.2%)	
University	67 (71.3%)	20 (60.6%)	
**Self-Perceived SES (Score)**	5.6 ± 1.4	5.0 ± 1.5	0.036
**Employment situation**			
Employee	27 (28.7%)	6 (18.2%)	0.368
Unemployed	8 (8.5%)	4 (12.1%)	
Independent	30 (31.9%)	11 (33.3%)	
Pensioner	29 (30.9%)	11 (33.3%)	
**Self-Perceived Financial Situation**			
Fairly good	7 (7.4%)	1 (3.0%)	0.417
Good	31 (33.0%)	12 (36.4%)	
Regular	52 (55.3%)	17 (51.5%)	
Very bad	4. (4.3%)	2 (6.1%)	
**Marital Status**			
Single	29 (30.9%)	10 (30.3%)	0.238
Married/Partnered	42 (44.7%)	16 (48.5%)	
Divorced	19 (20.2%)	3 (9.1%)	
Widower	4 (4.3%)	4 (12.1%)	

Abbreviations: DPP—diabetes prevention program; LD—liquid diet; SES—socioeconomic status.

**Table 3 nutrients-14-00939-t003:** Baseline evaluation—characteristics of all the participants for intervention.

	Total	Liquid Diet	Only DPP	*p*-Value
*n*	127	64	63	
Women *n* (%)	91 (71.7%)	49 (76.6%)	42 (66.7%)	0.216
Age (years)	52.1 ± 12.0	53.9 ± 12.8	50.3 ± 11.0	0.092
Height (cm)	160.1 ± 8.3	159.5 ± 8.5	160.7 ± 8.1	0.402
Weight (kg)	77.8 ± 13.5	76.8 ± 12.6	78.9 ± 14.3	0.395
Body mass index (kg/m^2^)	30.2 ± 3.7	30.1 ± 3.6	30.3 ± 3.7	0.734
Waist circumference (cm)	104.5 ± 8.8	104.6 ± 8.3	104.4 ± 9.4	0.942
Percentage of fat (%)	40.3 ± 7.8	40.9 ± 7.4	39.7 ± 8.2	0.380
Systolic blood pressure (mmHg)	127.2 ± 18.3	125.4 ± 16.6	129.1 ± 19.9	0.259
Diastolic blood pressure (mmHg)	74.2 ± 9.7	72.9 ± 8.5	75.6 ± 10.7	0.119
Fasting blood glucose (mg/dL)	86.3 ± 7.9	86.2 ± 8.1	86.3 ± 7.7	0.962
2-h blood glucose (mg/dL)	92.3 ± 24.4	94.5 ± 27.2	90.0 ± 21.1	0.296
Total cholesterol (mg/dL)	180.4 ± 42.9	179.8 ± 46.6	181.0 ± 39.2	0.874
LDL-c (mg/dL)	109.7 ± 32.8	107.3 ± 34.3	112.1 ± 31.3	0.413
VLDL-c (mg/dL)	25.8 ± 11.0	24.1 ± 9.3	27.4 ± 12.3	0.096
HDL-c (mg/dL)	41.1 ± 8.1	40.9 ± 8.8	41.3 ± 7.3	0.783
Triglycerides (mg/dL)	139.5 ± 94.9	141.8 ± 119.3	137.2 ± 61.7	0.783
EQ-5D	70.0 ± 15.9	70.7 ± 15.7	69.3 ± 16.2	0.617
Prediabetes (%)	7 (5.5%)	4 (6.3%)	3 (4.8%)	0.509
Hypertension (%)	69 (54.3%)	28 (43.8%)	41 (65.1%)	0.013
Anti-hypertensive medication (%)	25 (19.7%)	10 (15.6%)	15 (23.8%)	0.175
High cholesterol (%)	39 (30.7%)	21 (32.8%)	18 (28.6%)	0.373
High triglycerides (%)	35 (27.6%)	18 (28.1%)	17 (27.0%)	0.522
Statin use (%)	8 (6.3%)	4 (6.3%)	4 (6.3%)	0.632
**Social Determinants**				
**Education Level**				
Primary	13 (10.2%)	8 (12.5%)	5 (7.9%)	0.521
Secondary	27 (21.3%)	15 (23.4%)	12 (19.0%)	
University	87 (68.5%)	41 (64.1%)	46 (73.0%)	
**Self-Perceived SES (Score)**	5.5 ± 1.5	5.5 ± 1.4	5.4 ± 1.5	0.577
**Employment situation**				
Employee	33 (26.0%)	13 (20.3%)	20 (31.7%)	0.081
Unemployed	12 (9.4%)	5 (7.8%)	7 (11.1%)	
Independent	41 (32.3%)	18 (28.1%)	23 (36.5%)	
Pensioner	40 (31.5%)	27 (42.2%)	13 (20.6%)	
**Self-Perceived Personal Financial** **Situation**				
Fairly good	8 (6.3%)	6 (9.4%)	2 (3.2%)	0.081
Good	43 (33.9%)	15 (23.4%)	28 (44.4%)	
Regular	69 (54.3%)	38 (59.4%)	31 (49.2%)	
Very bad	6 (4.7%)	4 (6.3%)	2 (3.2%)	
**Marital Status**				
Single	39 (30.7%)	21 (32.8%)	18 (28.6%)	0.106
Married/Partnered	58 (45.7%)	23 (35.9%)	35 (55.6%)	
Divorced	22 (17.3%)	15 (23.4%)	7 (11.1%)	
Widower	8 (6.3%)	5 (7.8%)	3 (4.8%)	

Abbreviations: DPP—diabetes prevention program; LD—liquid diet; SES—socioeconomic status.

**Table 4 nutrients-14-00939-t004:** Modifiable components at the first phase of the intervention.

Variables	Intervention	Baseline (*n* = 94)	2 Months	Change	*p*-Value	≠	*p*-Value
Weight	LD	74.2 ± 10.6	71.3 ± 9.4	−2.9	<0.001	0.7	0.170
	DPP	79.1 ± 13.5	76.9 ± 13.8	−2.2	<0.001		
Body mass index (kg/m^2^)	LD	29.9 ± 3.5	28.8 ± 3.4	−1.2	<0.001	0.4	0.112
	DPP	30.6 ± 3.7	29.8 ± 3.8	−0.8	0.124		
Waist circumference (cm)	LD	103.0 ± 7.0	99.8 ± 6.3	−3.2	<0.001	1.4	0.113
	DPP	104.4 ± 9.1	102.6 ± 9.0	−1.8	<0.001		
Fat percentage (%)	LD	42.1 ± 7.5	40.7 ± 7.4	−1.4	<0.001	−0.5	0.323
	DPP	40.5 ± 8.4	38.6 ± 8.2	−1.9	<0.001		
Systolic blood pressure (mm/Hg)	LD	125.5 ± 16.5	115.0 ± 15.0	−10.5	<0.001	2.1	0.521
	DPP	130.4 ± 19.9	122.0 ± 16.2	−8.4	0.002		
Diastolic blood pressure (mm/Hg)	LD	72.7 ± 8.1	70.6 ± 10.1	−2.1	0.112	0.3	0.441
	DPP	75.7 ± 9.4	73.9 ± 8.8	−1.8	<0.001		
Fasting blood glucose	LD	87.0 ± 7.4	86.6 ± 8.5	−0.4	0.812	−2.0	0.320
	DPP	86.5 ± 8.0	84.1 ± 9.3	−2.4	0.113		

LD: liquid diet; DPP: diabetes prevention program.

## Data Availability

The data presented in this study are available on request from the corresponding author. The data are not publicly available due to privacy reasons.

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
