# Peer review of "Feasibility and Effectiveness of a Preventive Care Program during the Compound Humanitarian Crisis and COVID-19 Pandemic in Venezuela"

_nutrients, 2022, doi:10.3390/nu14050939_

Round 1

Reviewer 1 Report

The study presented is very interesting. It seems to have been extremely difficult to carry out in the existing circumstances (humanitarian crisis and epidemic). Implementing an intervention based on a liquid diet is difficult even when the diet replacements were provided without cost to participants, consisting of commercial nutritional formulas with instant preparation and known nutritional composition.

Specific comments and objections

1) It is necessary to include in the paper a flow diagram showing the design and conduct of the study.

2) I have great doubts about the use of the Tanita UM-081 device. The Tanita Body Fat Monitor UM-081 is intended for home use only and is not intended for professional use in hospitals or other medical facilities; it is not equipped with the quality standards. The % fat results obtained with this device are unreliable.  It should also be added that Tanita UM-081 does not measure visceral fat, which is key in assessing diabetes and cardiovascular risk. Adding this parameter would significantly enrich the results of the study.

3) Therefore, the section containing the limitations of the study should be expanded to include concerns related to the methodology of body composition assessment, or the % fat results should be removed from the article.

4). The reference list lacks numbering which makes it very difficult to review the paper.

Author Response

Dear Reviewer, 

Thanks for your comments. 

1- The flowchart with the design of the study was included in figure 1

2 and 3- We included the limitation of the device used in the limitations section

4- there was a problem when the file was uploaded, this problem should be corrected 

Reviewer 2 Report

In this manuscript entitled 'Feasibility and Effectiveness of a Preventive Care Program during the Compound Humanitarian Crisis and COVID-19 Pandemic in Venezuela. The HITS Diabetes with Prevention Study' the authors assess the implementation process of feasibility and fidelity, and the effectiveness in weight loss of using a hybridized approach with LD, medical nutrition therapy, and an adapted DPP vs. an adapted DPP alone. 
This pragmatic clinical trial compares the weight loss achieved with two lifestyle intervention programs in a community health center, the study design includes two phases: Phase 1 (the first two months of intervention - LD Group Step 1 vs. DPP Group) and Phase 2 (end of the sixth month when both groups finalize the intervention). 127 participants were randomized, and 94/127 participants (74.0%) completed the first phase. 
The authors worked on an interesting topic looking at a feasible and effective strategy to create weight loss with an improvement of cardiometabolic risk factors in a primary setting of Venezuela. The manuscript, in my opinion, is of good quality but has some limitations and issues to clarify.

Major Concerns
Material and Methods - Data analysis
This paragraph appears repetitive in some places, the same tests are mentioned several times or how the data are presented. A better summary can be made. More detailed information should be provided on correlation and regression analysis. 

Results
Comparisons at 2 months after the start of the intervention for the two arms under analysis are presented in Table 4. How was the overall difference assessed? In this analysis the baseline should always be taken into account by using the adjustment covariate in a regression model.

A scatterplot with the weight on the Y-axis and the number of sessions on the X-axis is presented in figure 2. The result should indicate how strong the linear correlation between these two variables is and not the R2 value which indicates the percentage of the variance in the dependent variable that the independent variables explain collectively. The statistic to be used should be pearson's r or even better spearman's rho if integer data (number of sessions) are used. It would also be interesting to evaluate the intraclass correlation.

Figure 3 shows the decrease in weight according to intervention arm and attendance level. I am afraid that single t-tests were carried out, but a two-way anova procedure (with correction for multiple tests) should be used for adequate error correction.

Author Response

Dear reviewer, thanks for your comments and suggestions. 

  1. Data analysis section: the paragraph was improved with detailed explanations.
  2. Table 4; the results were analyzed adjusting by their baseline values. 
  3. Figure 2 Scatterplot: the correlation between the variables was included
  4. Figure 3: the analysis was done with adjustments by baseline values of the subgroups.